# InvBench: Can LLMs Accelerate Program Verification with Invariant Synthesis?

## Abstract

Program verification relies on loop invariants, yet automatically discovering strong invariants remains a long-standing challenge. We introduce a principled framework for evaluating LLMs on invariant synthesis. Our approach uses a verifier-based decision procedure with a formal soundness guarantee and assesses not only correctness but also the speedup that invariants provide in verification. We evaluate 7 state-of-the-art LLMs, and existing LLM-based verifiers against the traditional solver UAutomizer. While LLM-based verifiers represent a promising direction, they do not yet offer a significant advantage over UAutomizer. Model capability also proves critical, as shown by sharp differences in speedups across models, and our benchmark remains an open challenge for current LLMs. Finally, we show that supervised fine-tuning and Best-of-N sampling can improve performance: fine-tuning on 3589 instances raises the percentage of speedup cases for Qwen3-Coder-480B from 8% to 29.2%, and Best-of-N sampling with N=16 improves Claude-sonnet-4 from 8.8% to 22.1%. We release our dataset for training and evaluation at `https://anonymous.4open.science/r/InvBench/`.

## 1 Introduction

Program verification aims to provide formal guarantees that software behaves as intended, with applications in many safety-critical domains (Fan et al., 2017; Luckcuck et al., 2019). A long-standing challenge in this area, studied for more than four decades, is the automatic discovery of loop invariants. In this work, we investigate whether large language models (LLMs) can accelerate program verification by generating useful loop invariants.

Loop invariants are conditions that hold before and after each loop iteration, and they are central to deductive program verification. To accelerate program verification, loop invariants must not only be correct but also sufficiently strong to prove the assertions. Generating correct invariants is relatively easy, since any universally true condition qualifies. However, only strong invariants can reduce verification effort and lead to a speedup. For example, in Figure 1, the invariant $x > 0$ is correct but not strong enough to prove the final assertion $x \neq 145$, whereas $x \equiv 3 \pmod 7$ is both correct and sufficiently strong.

Discovering such invariants is difficult and undecidable in general, which has motivated a long line of research. Traditional approaches include constraint solving (Colon et al., 2003; Gupta et al., 2009), dynamic analysis (Le et al., 2019), etc. Since invariant discovery is undecidable in general, researchers have tried a variety of learning-based methods (Li et al., 2017; Ezudheen et al., 2018). Building on this progression, the strong capabilities of LLMs in code generation and program reasoning (Austin et al., 2021; Chen et al., 2021; Wei et al., 2025b) naturally motivate a systematic evaluation of their potential for invariant discovery.

Pei et al. (2023) is the first work to evaluate the capabilities of LLMs in invariant generation. However, their methodology considers only correctness and does not assess how strong the generated invariants are. As a result, LLMs may generate correct invariants that perform well under their evaluation metric but provide no benefit for accelerating the verification process in real-world settings. Furthermore, their notion of correctness is not based on formal verification. Instead, it is determined by direct comparison with invariants generated by an existing tool, namely Daikon (Ernst et al., 2007). Daikon is a dynamic analysis tool whose invariants are not guaranteed to be sound, since they are inferred from observed test executions rather than proven across all possible executions. As a result, the

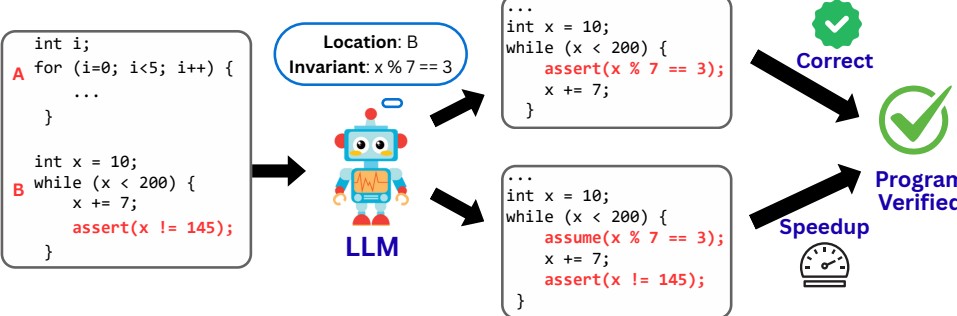

Figure 1: Illustration of InvBench's evaluation pipeline. The LLM proposes an invariant by specifying a program location and predicate (e.g., location B with `x % 7 == 3`). The verification procedure then incorporates this invariant to prove the property `x != 145` using two verifier queries, and we measure the resulting speedup relative to a baseline without LLM assistance.

ground-truth invariants themselves may be incorrect. Moreover, directly comparing LLM-generated invariants with Daikon's output can lead to rejecting many correct invariants. For example, if an LLM proposes $a > 0$ for an integer $a$ while Daikon reports $a \geq 1$, the evaluation in Pei et al. (2023) would incorrectly classify the LLM's result as wrong, even though the two are equivalent. Hence, prior work's evaluation methodology cannot reliably capture the correctness of LLM-generated invariants, let alone how useful they are for verification.

A series of follow-up works inspired by Pei et al. (2023) have proposed LLM-based verifiers (Wu et al., 2024b;a; Kamath et al., 2024). Instead of evaluating LLMs in isolation, these efforts develop verification frameworks powered by LLMs. However, each work introduces a custom dataset and reports results only on it, making cross-comparison difficult and raising concerns about generalization.

Our work, InvBench, introduces a principled methodology for evaluating the capabilities of LLMs in invariant synthesis. Instead of checking whether LLMs reproduce invariants discovered by other tools, we employ a verifier-based decision procedure that directly determines the correctness of LLM-generated invariants, and we prove this procedure to be sound. We formalize the methodology as a proof calculus, providing a rigorous foundation for invariant evaluation. Unlike previous work that focuses on optimizing verification performance by designing tailored interactive protocols between solvers and LLMs (Chakraborty et al., 2023; Wu et al., 2024b; Kamath et al., 2024; Wu et al., 2024a), our goal is to develop a simple, *first-order* verification procedure suitable for evaluating the invariant generation capability of LLMs. Our formalization provides such a framework, offering a general solution for invariant evaluation.

Since the purpose of invariant synthesis is to accelerate verification, invariants that are too weak to aid verification or too difficult to verify as correct offer little practical value. To capture this, we evaluate the invariants by measuring the speedup they provide in the overall verification process.

To support comparison across solvers and LLMs, we construct a dataset of 226 instances derived from the most recent edition of the software verification competition SV-Comp (Beyer & Strejček, 2025) and use it to evaluate multiple LLM-based verifiers on this common benchmark. In addition, we assess the state-of-the-art traditional (i.e., non-LLM-based) verifier UAutomizer (Schüssele et al., 2024) both on our dataset and on each of the custom datasets introduced in prior work (Wu et al., 2024b; Kamath et al., 2024; Wu et al., 2024a).

UAutomizer consistently outperforms prior LLM-based verifiers on both our dataset and their custom datasets across all settings, with the exception of LEMUR (Wu et al., 2024b), which was specifically designed for problems that UAutomizer fails to solve. These results suggest that while LLM-based verifiers are a promising direction, existing approaches do not yet offer a significant advantage over non-LLM-based approaches in general. Our findings also indicate that the effectiveness of LLM-based verifiers is strongly determined by the underlying symbolic solver they rely on.

In addition, we evaluate 7 state-of-the-art LLMs. To improve models' capabilities, we construct a fine-tuning dataset of 3589 instances and show that training on this dataset raises the percentage of

speedup cases for Qwen3-Coder-480B from 8% to 29.2%. Similarly, Best-of-N sampling with N = 16 improves Claude-sonnet-4 from 8.8% to 22.1%. Our methods, despite the simplicity, establish a new performance baseline. Our decision procedure enables LLMs to outperform UAutomizer, whereas prior LLM-based verifiers with significantly more complex designs rarely do so.

In summary, our contributions are as follows:

- We propose a verification procedure for evaluating the invariant generation capabilities of LLMs, assessing both correctness and their effectiveness in accelerating verification.
- We evaluate 7 state-of-the-art LLMs, and provide comparisons between existing LLM-based verifiers and the state-of-the-art non-LLM-based solver UAutomizer.
- We construct a dataset of 3589 instances for training. We demonstrate that both supervised fine-tuning and Best-of-N sampling can easily improve model performance in accelerating verification, establishing a new performance baseline.

## 2 RELATED WORK

**Traditional Methods for Program Invariant Generation.**   A long line of research has explored invariant synthesis using traditional techniques without machine learning, including model checking (Flanagan & Qadeer, 2002; Lahiri & Bryant, 2007; Hojjat & Rümmer, 2018; Vediramana Krishnan et al., 2024), abstract interpretation (Karr, 1976; Cousot & Cousot, 1977; Cousot & Halbwachs, 1978; Cousot & Cousot, 1979), constraint solving (Gulwani et al., 2009; Gupta et al., 2009), Craig interpolation (Jhala & McMillan, 2006; McMillan, 2010), and syntax-guided synthesis (Fedyukovich & Bodík, 2018). Prior work evaluating LLM-generated invariants (Pei et al., 2023) has relied on Daikon (Ernst et al., 2007), a tool for dynamic invariant detection (Echenim et al., 2019; Le et al., 2019). Daikon executes the program, observes runtime values, and reports properties that consistently hold over the observed executions. However, such invariants may fail to generalize to all possible executions, thereby compromising soundness. Our approach instead employs a verifier-based decision procedure relying on UAutomizer (Schüssele et al., 2024) that ensures soundness.

**Learning-Based Method for Invariant Generation.**   Machine learning based techniques have been widely adopted in invariant synthesis, including decision tree (Garg et al., 2014; 2016; Ezudheen et al., 2018; Riley & Fedyukovich, 2022; Xu et al., 2020), support vector machine (Li et al., 2017; Sharma et al., 2012), reinforcement learning (Si et al., 2018; Yu et al., 2023), and others (Sharma et al., 2013; Ryan et al., 2019; Yao et al., 2020). More recently, large language models have demonstrated strong capabilities in reasoning about code and logic (Wei et al., 2025a;b;c), giving rise to a series of work that explore using LLMs for finding invariants. Pei et al. (2023) is the first pioneering work that evaluates LLMs' capabilities in finding invariants, but it is not a sound evaluation. Various techniques have been proposed to couple LLMs with symbolic solvers, including ranking LLM-generated invariants (Chakraborty et al., 2023), the "query-filter-reassemble" strategy of LaM4Inv (Wu et al., 2024a), the back-tracking algorithm in LEMUR (Wu et al., 2024b), and Loopy's integration of the classic Houdini algorithm (Kamath et al., 2024). On the dataset side, Liu et al. (2024) introduces a rule-based method for constructing a fine-tuning corpus, which differs from our verifier-based approach. In contrast, our work provides a simple and sound evaluation procedure for assessing LLM-generated invariants and investigates how both fine-tuning and Best-of-N sampling can enhance LLM performance in invariant synthesis.

## 3 METHOD

### 3.1 PRELIMINARY

We formalize the task of loop invariant synthesis using standard Hoare logic (Hoare, 1969). A Hoare triple $\{P\} S \{Q\}$ specifies that if the precondition $P$ holds before executing a statement $S$, then the postcondition $Q$ will hold after its execution. In the context of loops, an invariant $I$ is a logical proposition that summarizes the state of the program at each iteration, and it is the key to proving the validity of Hoare triples involving loops. For a loop of the form while $B$ do $S$, the goal of invariant synthesis is to identify a loop invariant $I$ that satisfies the following inference rule:

$$\frac{P \Rightarrow I \quad \{I \wedge B\} \, S \, \{I\} \quad I \wedge \neg B \Rightarrow Q}{\{P\} \, \texttt{while} \, B \, \texttt{do} \, S \, \{Q\}}$$

Here, $P$ is the precondition, $Q$ is the postcondition, $B$ is the loop condition, and $S$ is the loop body. Intuitively, the inference rule requires that the invariant $I$ holds at the beginning of the loop ($P \Rightarrow I$), is preserved across every iteration of the loop body ($\{I \wedge B\} \, S \, \{I\}$), and upon termination ensures the postcondition ($I \wedge \neg B \Rightarrow Q$).

Invariant synthesis amounts to generating a logical summary $I$ that is both *correct*, meaning it can be verified, and *strong*, meaning it enables verification of the final assertion. Weak but correct invariants contribute little, leaving most of the reasoning to the verifier, whereas strong invariants narrow the search space of program states, reduce solver effort, and yield substantial speedups.

## 3.2 VERIFIER-BASED DECISION PROCEDURE

We formalize our verifier-based procedure for assessing candidate invariants. Let $P$ denote a program. A *property* is written as $p = \langle \varphi, \ell \rangle$, where $\varphi$ is a state predicate and $\ell$ is a program location. For a finite set $A$ of properties, let $\mathrm{Asm}(P, A)$ be the program obtained from $P$ by inserting $\texttt{assume}$ statements for all elements of $A$. An execution of $\mathrm{Asm}(P, A)$ that reaches a location where an assumption is violated terminates immediately. We write $P \models_A p$ to indicate that all executions of $\mathrm{Asm}(P, A)$ satisfy the assertion $p$. The notation $P \models p$ abbreviates $P \models_\varnothing p$. Since assumptions restrict behaviors, if $P \not\models_A p$ for some $A$, then necessarily $P \not\models p$.

We assume access to a verifier

$$V(P, A, p) \in \{\mathbf{T}, \mathbf{F}, \mathbf{U}\},$$

which returns either $\mathbf{T}$ (proved), $\mathbf{F}$ (refuted), or $\mathbf{U}$ (inconclusive). The verifier is required to be sound on conclusive outcomes:

$$V(P, A, p) = \mathbf{T} \;\Rightarrow\; P \models_A p, \qquad V(P, A, p) = \mathbf{F} \;\Rightarrow\; P \not\models_A p.$$

No completeness is assumed for $\mathbf{U}$, which may arise from timeouts or incompleteness of the underlying verifier.

The verification task specifies a target property $p^\star = \langle \varphi^\star, \ell^\star \rangle$. Given $P$ and $p^\star$, a large language model proposes a *candidate invariant* $q = \langle \psi, \ell \rangle$, typically at a loop header. To evaluate the utility of $q$, the procedure issues two verifier queries:

$$d_a := V(P, \varnothing, q) \quad \text{(checking whether } q \text{ is a correct predicate)},$$

$$d_b := V(P, \{q\}, p^\star) \quad \text{(checking whether the target holds under the assumption } q\text{)}.$$

An example of the two verifier queries are given in Figure 1. The outcome of the procedure is expressed as a judgment

$$P \;\Rightarrow\; \langle p^\star, q \rangle \;\Downarrow\; d \qquad \text{with} \quad d \in \{\mathbf{T}, \mathbf{F}, \mathbf{U}\}.$$

The interpretation is as follows: if the judgment yields $\mathbf{T}$, then $p^\star$ is established on $P$; if it yields $\mathbf{F}$, then $p^\star$ is refuted; and if it yields $\mathbf{U}$, the attempt is inconclusive.

The inference rules defining this judgment are given below. Each rule specifies one possible derivation of the outcome, depending only on the responses of the verifier.

$$\frac{V(P, \{q\}, p^\star) = \mathbf{F}}{P \;\Rightarrow\; \langle p^\star, q \rangle \;\Downarrow\; \mathbf{F}} \text{ (DEC-FALSE)}$$

$$\frac{V(P, \varnothing, q) = \mathbf{T} \qquad V(P, \{q\}, p^\star) = d \qquad d \neq \mathbf{F}}{P \;\Rightarrow\; \langle p^\star, q \rangle \;\Downarrow\; d} \text{ (DEC-PROP)}$$

$$\frac{V(P, \varnothing, q) \neq \mathbf{T} \qquad V(P, \{q\}, p^\star) \neq \mathbf{F}}{P \;\Rightarrow\; \langle p^\star, q \rangle \;\Downarrow\; \mathbf{U}} \text{ (DEC-U)}$$

Rule DEC-FALSE captures short-circuit refutation: if the goal fails even in the restricted program $\text{Asm}(P, \{q\})$, then it is false on the original program $P$. Rule DEC-PROP implements the prove-then-use strategy: once the candidate invariant $q$ is established, the outcome is exactly the verifier's answer on the goal under the assumption $q$, restricted to $d \in \{\mathbf{T}, \mathbf{U}\}$ so as not to overlap with DEC-FALSE. Rule DEC-$\mathbf{U}$ gives explicit conditions for inconclusiveness: the goal is not refuted under $q$ and $q$ is not established as an invariant.

**Theorem** (Decision Soundness). *If $P \Rightarrow \langle p^\star, q \rangle \Downarrow \mathbf{T}$ is derivable, then $P \models p^\star$. If $P \Rightarrow \langle p^\star, q \rangle \Downarrow \mathbf{F}$ is derivable, then $P \not\models p^\star$.*

The proof is provided in Section A.1. This theorem establishes that whenever the calculus derives a conclusive outcome, that outcome is correct. The inconclusive case $\mathbf{U}$ is deliberately conservative: it makes no claim about the truth or falsity of the property and safely captures verifier incompleteness or timeouts.

### 3.3    IMPLEMENTATION

We describe the implementation of our verifier-based evaluation framework. Given a program $P$ and a target property $p^\star$, the system must generate candidate invariants $q$ and evaluate them according to the decision procedure. When proposing an invariant $q = \langle \psi, \ell \rangle$, the model selects a program location $\ell$ and supplies the corresponding predicate $\psi$.

**Syntactic Validation.**    Before invoking the verifier, we apply syntactic checks to the generated predicate $\psi$. These checks ensure that $\psi$ can be safely interpreted as a state predicate and that its insertion as an assumption does not alter the program state. For instance, expressions that update variables (e.g., `a += 1`) are rejected. Only Boolean conditions over the program state are accepted.

**Parallel Evaluation.**    For each candidate $q$, the procedure issues two verifier queries, namely $d_a = V(P, \varnothing, q)$ to check whether $q$ is an invariant and $d_b = V(P, \{q\}, p^\star)$ to check whether the target holds under the assumption $q$. These queries are executed in parallel in our implementation, which reduces latency and enables speedup when the proposed invariant is useful for verification. The final outcome is then derived exactly according to the decision calculus.

### 3.4    SUPERVISED FINE-TUNING AND BEST-OF-N SAMPLING

We perform supervised fine-tuning using LoRA (Hu et al., 2022). Below, we discuss how we construct our dataset for fine-tuning and the way we perform Best-of-N sampling.

**Synthetic Dataset Generation.**    To construct the synthetic dataset, we prompt GPT-4o using the template in Appendix A.3. The template takes three seed programs as examples and instructs the model to synthesize a new C program that is compilable and contains both loops and assertions. To obtain a diverse and large pool of candidates, we repeatedly invoke the model with different seed programs. To avoid data leakage, these seed programs are randomly drawn from the SV-COMP pool (Beyer & Strejček, 2025) that is *disjoint* from our evaluation set. Although the prompt requests compilable programs with loops and assertions, the LLM-generated programs may fail to compile, include assertions that do not hold, and contain multiple assertions. For programs with multiple assertions, we split them into separate instances, each retaining only a single assertion while preserving all loop structures (ensuring at least one loop per instance). We then run UAutomizer on every program and discard any instance that is non-compilable or whose assertion is invalid. This filtering step ensures the quality of the dataset, resulting in 3589 synthetic programs.

**Extract Invariants Generated from UAutomizer.**    When running UAutomizer to prove the assertions in the synthetic programs, the tool also emits the invariants it discovers. From its output, we extract the loop invariants. Each extracted invariant includes its program location and its predicate. Although each program contains exactly one assertion, it may include multiple loops, so a single program can yield multiple loop invariants, all associated with the same assertion. We pair each program with each of its corresponding loop invariants to form our training dataset. We show an example invariant generated from UAutomizer in Appendix A.4.

**Best-of-N Sampling.** Best-of-N sampling is an inference-time strategy in which multiple candidate programs are generated, and the most effective one is selected, a technique shown to improve performance on code-generation tasks (Ehrlich et al., 2025). In our setting, the best candidate is the invariant that yields the largest speedup, i.e., the one whose decision procedure finishes earliest. As described in Section 3.2, evaluating a single candidate requires two verifier queries issued in parallel; therefore, Best-of-N sampling evaluates 2N verifier queries concurrently.

## 4 EXPERIMENTAL SETUP

**Dataset from SV-COMP.** We construct our benchmark from SV-COMP (Beyer & Strejček, 2025), a standard competition in software verification, focusing on problems that require loop invariant synthesis. We collect a pool of 899 instances and run the state-of-the-art non-LLM verifier UAutomizer (Schüssele et al., 2024) with a 600-second timeout to record the solving time for each. Based on this runtime, we classify instances into an *easy split* (solved within 30 seconds) and a *hard split* (solved between 30 and 600 seconds). From this pool, we randomly sample 113 problems from each split, resulting in 226 instances in the final evaluation set. The selection process is fully automated with no manual cherry-picking.

**Synthetic Dataset.** We construct a synthetic corpus of verification problems using GPT-4o. Seed programs supplied in the prompt are selected to ensure no overlap with the 226 instances in the evaluation set. Each synthetic program is analyzed by UAutomizer. The invariants extracted from UAutomizer's execution output form our fine-tuning dataset, which contains 3589 problems paired with their invariants.

| Dataset | Split | Avg. #Lines | #Instances |
|---------|-------|-------------|------------|
| Evaluation | Easy | 51 | 113 |
| | Hard | 62 | 113 |
| Training | – | 42 | 3589 |

Table 1: Dataset statistics of InvBench.

**Metrics.** All speedup-related measurements are reported relative to the state-of-the-art non-LLM-based solver UAutomizer (Schüssele et al., 2024), which serves as the baseline solver. We evaluate LLMs along two dimensions: the correctness of the generated invariants and the performance improvements they provide. Correctness is judged by the decision procedure formalized in Section 3 with a timeout set to the problem's original solving time by UAutomizer. For comparisons between UAutomizer and other tools, we report the number of solved instances under varying time budgets. We include the model's token generation time in all evaluations.

**Models.** We benchmark Claude models from Anthropic, GPT models from OpenAI, and the Qwen family of models (Hui et al., 2024; Yang et al., 2025).

**Hardware and OS.** Experiments were conducted on a server with Intel Xeon Platinum 8275CL CPUs (96 cores), 8 NVIDIA A100 GPUs, and 1.1 TB of memory, running Ubuntu 22.04.

## 5 RESULTS

### 5.1 RESULTS OF LLMS

We report the performance of different LLMs on the easy and hard splits of InvBench in Table 2 and Table 3, respectively. On the easy split shown in Table 2, o3 achieves the strongest results, with a 1.37× speedup over UAutomizer on 28.3% of problems and an overall 1.09× average speedup. These results indicate that while state-of-the-art LLMs generally struggle to synthesize correct or sufficiently strong invariants to accelerate program verification, the strongest model o3 can still yield non-trivial improvements on a meaningful fraction of problems.

Table 3 reports results on the hard split of InvBench. In this setting, LLMs show only negligible improvement over UAutomizer. The only noteworthy exception is gpt-oss-120b, which produces an

| Model | % Correct Invariant | % Speedup | Speedup$_{>1}$ | Speedup$_{all}$ |
|---|---|---|---|---|
| Qwen2.5-72B | 4.4% | 0.9% | 1.20× | 1.00× |
| gpt-oss-120b | 8.8% | 5.3% | 1.10× | 1.01× |
| claude-sonnet-4 | 15.0% | 8.8% | 1.06× | 1.01× |
| Qwen3-Coder-480B | 14.2% | 8.0% | 1.09× | 1.01× |
| claude-opus-4.1 | 18.6% | 8.8% | 1.23× | 1.02× |
| gpt-5 | 37.2% | 26.5% | 1.24× | 1.06× |
| o3 | 39.8% | 28.3% | 1.37× | 1.09× |

Table 2: **InvBench-Easy**: Results on the InvBench dataset (easy split) across models. We report the percentage of instances with verified-correct invariants, the percentage of instances achieving speedup greater than 1, the average speedup over those cases (Speedup$_{>1}$), and the average speedup across all instances with non-speedups counted as 1 (Speedup$_{all}$).

| Model | % Correct Invariant | % Speedup | Speedup$_{>1}$ | Speedup$_{all}$ |
|---|---|---|---|---|
| gpt-5 | 11.5% | 0% | 1.00× | 1.00× |
| Qwen2.5-72B | 12.4% | 0% | 1.00× | 1.00× |
| o3 | 29.2% | 0% | 1.00× | 1.00× |
| Qwen3-Coder-480B | 15.9% | 0% | 1.00× | 1.00× |
| claude-sonnet-4 | 13.3% | 0.9% | 3.35× | 1.01× |
| claude-opus-4.1 | 14.2% | 0.9% | 2.96× | 1.02× |
| gpt-oss-120b | 11.5% | 0.9% | 29.57× | 1.03× |

Table 3: **InvBench-Hard**: Results on the InvBench dataset (hard split) across models.

invariant for one problem that delivers a 29.57× speedup. However, overall, such speedup cases are exceedingly rare.

There are three main takeaways from the results. First, generating strong invariants that yield performance speedups is substantially more difficult than merely producing correct invariants, as reflected in the large gap between the percentage of correct invariants and the percentage of speedups. Second, model capability is a key factor, as demonstrated by the sharp differences in speedups reported in Table 2. Third, LLMs remain far from fully addressing the task of invariant synthesis, leaving considerable room for future progress, as shown in Table 3.

## 5.2 Results of Verifiers on InvBench

We compare the state-of-the-art non-LLM-based tool UAutomizer (Schüssele et al., 2024) with other LLM-based verifiers on both the easy split and the hard split of InvBench.

On the easy split of InvBench, where UAutomizer solves all 113 instances within 30 seconds, it clearly surpasses all LLM-based verifiers. LaM4Inv (Wu et al., 2024a) and Loopy (Kamath et al., 2023) fail to solve any instance within this time limit, while LEMUR (Wu et al., 2024b) solves only 55 instances. These results indicate that none of the LLM-based verifiers provide any advantage over UAutomizer on the easy split.

Figure 2 presents the comparison on the hard split of InvBench. UAutomizer still delivers the best performance across almost all timeouts. UAutomizer consistently and significantly outperforms LaM4Inv and Loopy. In particular, LaM4Inv and Loopy solve fewer than half as many instances as UAutomizer across all timeouts. Notably, within 10 seconds, LEMUR solves more problems than UAutomizer, suggesting that it can accelerate some of the instances. Nevertheless, UAutomizer still outperforms LEMUR with longer timeouts, indicating that although LEMUR can offer early solving advantages on certain challenging instances, its overall capability is still limited compared to UAutomizer.

Given the consistently poor performance of LaM4Inv, which often either fails or times out, we conducted a manual investigation. Our analysis shows that LaM4Inv fails on some examples when applied to external datasets beyond those used in its custom evaluation. However, the preprocessing steps or manual annotations required by the tool are not documented. The official repository does not

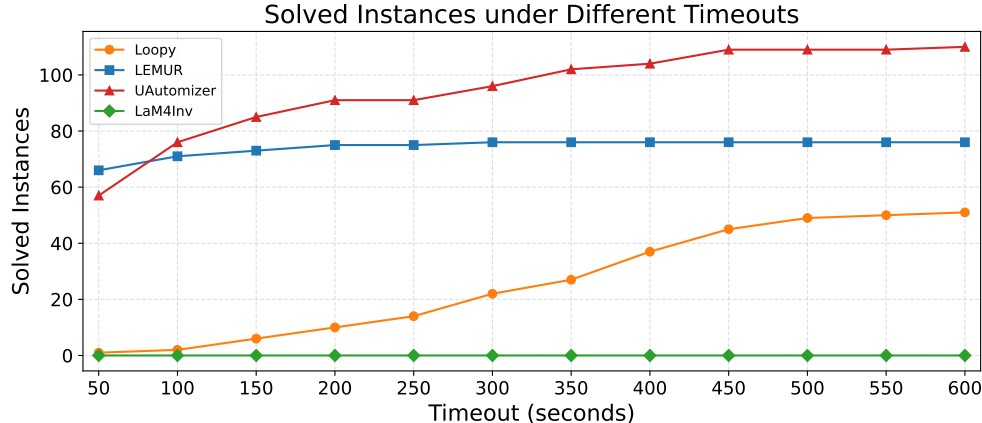

Figure 2: Comparison of solved instances on InvBench-Hard by different verifiers.

| Verifier | Total Instances | Solved Instances under Different Timeouts | | | |
|---|---|---|---|---|---|
| | | 10s | 100s | 300s | 600s |
| LaM4Inv (Wu et al., 2024a) | 316 | 144 | 286 | 295 | 299 |
| UAutomizer | 316 | 299 | 299 | 299 | 299 |
| Loopy (Kamath et al., 2023) | 469 | 0 | 133 | 353 | 403 |
| UAutomizer | 469 | 372 | 403 | 411 | 413 |
| LEMUR (Wu et al., 2024b) | 47 | 2 | 8 | 16 | 19 |
| UAutomizer | 47 | 0 | 0 | 0 | 0 |

Table 4: Comparison of prior LLM-based verifiers and UAutomizer on their own custom dataset under different timeout budgets.

provide preprocessing scripts, and our attempt to contact the authors has not received a response. For a direct comparison between UAutomizer and LaM4Inv, we also refer readers to Section 5.3, where we evaluate UAutomizer on the dataset released by LaM4Inv.

## 5.3 RESULTS OF UAUTOMIZER ON PRIOR DATASETS

As shown in Table 4, UAutomizer, the state-of-the-art non-LLM-based verifier, consistently solves more instances than LaM4Inv (Wu et al., 2024a) and Loopy (Kamath et al., 2023), two representative LLM-based tools, on their respective custom datasets. This highlights that prior work omitted an important baseline comparison against UAutomizer.

LEMUR (Wu et al., 2024b) is the only tool that surpasses UAutomizer. However, this is largely explained by its dataset construction: LEMUR evaluates only on problems that UAutomizer cannot solve within 600 seconds. To assess whether this advantage generalizes beyond its own curated benchmark, we conduct an additional analysis presented in Section 6.

While LaM4Inv and Loopy consistently underperform UAutomizer in terms of the total number of solved instances across time budgets, they nevertheless offer complementary benefits, which we discuss in Section A.2.

We further investigate the distributional differences between datasets. At the 600-second timeout, the performance of prior LLM-based verifiers approaches that of UAutomizer, with LaM4Inv even matching it exactly. In contrast, on InvBench (see Section 5.2), all LLM-based verifiers perform poorly at 600 seconds, suggesting a substantial shift in distribution compared to the custom datasets used in prior work. Our analysis confirms this: programs in InvBench are significantly longer, averaging 62 lines of code in the hard split, compared to only 22 for LaM4Inv, 23 for LEMUR, and 27 for Loopy. Manual inspection shows that InvBench contains features such as multiple loops,

functions, arrays, and pointers, which are largely absent from prior datasets. This makes InvBench a more challenging and realistic benchmark that better distinguishes solver performance.

We also note that different LLM-based verification frameworks are built on different base solvers: LEMUR is built on UAutomizer, Loopy on Frama-C (Cuoq et al., 2012), and LaM4Inv on ES-BMC (Gadelha et al., 2018). Given the strength of UAutomizer, we believe future work should place greater emphasis on developing LLM-based verifiers atop state-of-the-art solvers such as UAutomizer and ensure that comparisons against it are not omitted.

## 5.4 FAILURE MODE ANALYSIS

We conducted a detailed breakdown analysis to understand the primary failure modes behind the lack of speedups. For each model, we categorize failures into four types: 1) Incorrect Invariant: the candidate invariant is refuted; 2) Assume Timeout: verifying the invariant itself times out; 3) Assert Timeout: the invariant is verified, but verifying the final assertion under that invariant times out; 4) Assume + Assert Timeout: both checks time out. The table below summarizes this breakdown for the top three models on the easy split, showing the number of instances falling into each failure mode.

| Model | Incorrect | Assume Timeout | Assert Timeout | Assume + Assert Timeout |
|---|---|---|---|---|
| claude-opus-4.1 | 10 | 13 | 22 | 58 |
| o3 | 29 | 17 | 19 | 17 |
| gpt-5 | 20 | 4 | 18 | 65 |

Table 5: Failure mode breakdown on InvBench-Easy.

While a portion of failures stems from incorrect invariants, the more fundamental issue is that the invariants generated by LLMs rarely decompose the verification task into strictly easier subgoals. As a result, both the solver queries frequently time out, as reflected in the large number of cases in the "Assume + Assert Timeout" category. This suggests that current models lack an understanding of what makes a verification query easy or difficult for symbolic solvers.

Future work should explore strategies that help models internalize or predict solver difficulty, such as training reward models, so that they can propose invariants that genuinely simplify the verification task rather than inadvertently increasing solver burden.

## 5.5 EFFECTIVENESS OF FINE-TUNING

We perform supervised fine-tuning using LoRA (Hu et al., 2022) on the training set for 3 epochs.

Table 6 reports results on the easy split of InvBench. Fine-tuning leads to substantial gains in both invariant correctness and runtime speedup. Qwen3-Coder-480B shows the most pronounced improvements: the proportion of correct invariants increases from 14.2% in the base model to 40.7%, and the percentage of instances with speedups rises from 8% to 29.2%. For Qwen2.5-72B, the conditional speedup decreases after fine-tuning, but this is because the baseline conditional speedup was driven by a single case, whereas fine-tuning yields a larger number of cases with speedups. We also note that on the hard split, neither of the fine-tuned models shows improvement, demonstrating that our benchmark contains unsolved challenges.

| Model | % Correct Invariant | % Speedup | Speedup$_{>1}$ | Speedup$_{all}$ |
|---|---|---|---|---|
| Qwen2.5-72B (base) | 4.4% | 0.9% | 1.20× | 1.00× |
| Qwen2.5-72B (fine-tuned) | 32.7% | 26.5% | 1.13× | 1.03× |
| Qwen3-Coder-480B (base) | 14.2% | 8.0% | 1.09× | 1.01× |
| Qwen3-Coder-480B (fine-tuned) | 40.7% | 29.2% | 1.29× | 1.08× |

Table 6: Improvement of supervised fine-tuning on InvBench-Easy.

| Model | N | % Correct Invariant | % Speedup | Speedup$_{>1}$ | Speedup$_{all}$ |
|---|---|---|---|---|---|
| claude-opus-4.1 | 1 | 18.6% | 8.8% | 1.23× | 1.02× |
| claude-opus-4.1 | 16 | 37.2% | 23.0% | 1.15× | 1.03× |
| claude-sonnet-4 | 1 | 15.0% | 8.8% | 1.06× | 1.01× |
| claude-sonnet-4 | 16 | 36.3% | 22.1% | 1.20× | 1.04× |
| o3 | 1 | 39.8% | 28.3% | 1.37× | 1.09× |
| o3 | 16 | 50.4% | 35.4% | 1.39× | 1.12× |

Table 7: Results of Best-of-N sampling (N = 16) on InvBench-Easy.

## 5.6 EFFECTIVENESS OF BEST-OF-N SAMPLING

We also evaluated whether repeated sampling improves LLM performance in invariant synthesis. For each problem, we generated 16 samples at a temperature of 0.7, removed duplicate invariants, and verified all candidates in parallel. Table 7 summarizes the results on InvBench's easy split, demonstrating that repeated sampling leads to consistent gains. For example, o3 improves from 39.8% to 50.4% on correctness, while claude-opus-4.1 improves from 18.6% to 37.2%.

On the hard split, for claude-opus-4.1, with 16 samples, the percentage of instances with correct invariants increases from 14.2% to 15.9%, the percentage of instances with speedup from 0.9% to 2.7%, and the overall average speedup from 1.02× to 1.03×. Similarly, for claude-sonnet-4, the percentage of instances with correct invariants increases from 13.3% to 20.4%, the percentage of instances with speedup from 0.9% to 2.7%, conditional average speedup from 3.35× to 6.12×, and the overall average speedup from 1.01× to 1.07×. These improvements suggest that best-of-N sampling is a promising technique for LLMs to improve invariant synthesis performance.

## 6 DISCUSSION

**Performance Gains with Ground-Truth Invariants.** To quantify the potential speedup achievable when providing correct and strong invariants to solvers, we extract the invariants identified by UAutomizer and then measure the resulting speedup when they are supplied to it. On a random sample of 100 problems from our training set, we observe an overall average speedup of 1.86×. This indicates that the invariants discovered by UAutomizer are sufficiently strong for acceleration.

**Generalizability of LEMUR.** LEMUR (Wu et al., 2024b) reports the best results on its custom dataset, as it targets instances that UAutomizer cannot solve within 600 seconds. To test whether this advantage generalizes, we sampled 50 unsolved instances and found that LEMUR solved 12 within the same timeout of 600 seconds. This suggests that LEMUR's gains are not solely due to benchmark design but reflect a genuine advantage on harder problems.

**Inference Overhead of Models.** Our evaluation includes LLM serving time as a realistic measure of end-to-end performance. Since the goal is to accelerate verification, inference overhead must be considered alongside solver runtime. Future research may explore how to balance the quality of generated invariants with the inference cost of producing them.

## 7 CONCLUSION

This work introduced InvBench, a principled framework for evaluating the capabilities of LLMs in invariant synthesis. Our approach employs a verifier-based decision procedure with a formal soundness guarantee and assesses not only correctness but also the speedups that invariants contribute to program verification. Using a benchmark of 226 instances, we conducted a comparison across state-of-the-art LLMs, existing LLM-based verifiers, and the traditional solver UAutomizer. The results show that although LLM-based verifiers represent a promising direction, they do not yet offer significant advantages over non-LLM-based approaches. Model capability proves to be a critical factor, and our benchmark remains an open challenge for current LLMs. At the same time, we demonstrated that supervised fine-tuning and Best-of-N sampling can improve model performance in accelerating verification.

LLM USAGE

LLMs are the subject of this study. We additionally used them for polishing the writing.

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

# A APPENDIX

## A.1 PROOF

**Theorem** (Decision Soundness). *If $P \Rightarrow \langle p^\star, q \rangle \Downarrow \mathbf{T}$ is derivable, then $P \models p^\star$. If $P \Rightarrow \langle p^\star, q \rangle \Downarrow \mathbf{F}$ is derivable, then $P \not\models p^\star$.*

*Proof.* For the $\mathbf{T}$ case, the final rule must be DEC-PROP with $d = \mathbf{T}$. The premises yield $V(P, \varnothing, q) = \mathbf{T}$ and $V(P, \{q\}, p^\star) = \mathbf{T}$. By verifier soundness, $P \models q$ and $P \models_{\{q\}} p^\star$. Since $q$ holds on all executions of $P$, introducing the assumption $\{q\}$ does not remove executions relevant to $p^\star$; thus $P \models p^\star$.

For the $\mathbf{F}$ case, the final rule must be DEC-FALSE. Its premise $V(P, \{q\}, p^\star) = \mathbf{F}$ implies $P \not\models_{\{q\}} p^\star$ by soundness. Assumptions restrict behaviors; hence, a violation under assumptions entails a violation without them, yielding $P \not\models p^\star$. $\qquad\square$

## A.2 COMPLEMENTARY RESULTS OF LLM-BASED SOLVERS

As shown in Table A1, all three LLM-based tools solve instances that remain unsolved by UAutomizer within the 600-second budget. LaM4Inv and Loopy each contribute additional solved cases, and LEMUR is able to handle 19 problems that UAutomizer cannot solve at all, underscoring the complementary strengths of LLM-based approaches.

| Verifier | Solved Instances | $\Delta$ Solved |
|----------|------------------|-----------------|
| LaM4Inv  | 299              | 13              |
| Loopy    | 403              | 40              |
| LEMUR    | 19               | 19              |

Table A1: LLM-based verifiers complement UAutomizer by solving problems beyond its reach. We report the total numbers of solved problems, and "$\Delta$ Solved" is the number of instances uniquely solved that are unsolved by UAutomizer.

## A.3 PROMPT TEMPLATE USED FOR TRAINING DATASET GENERATION

Table A2 shows the prompt template used for synthesizing training programs from seed programs.

## A.4 AN EXAMPLE FROM THE FINE-TUNING DATASET

Figure A1 shows an example from the fine-tuning dataset with the program to the UAutomizer and the generated loop invariant.

The loop invariant holds at Line 6 (the beginning of the loop). It is a disjunction of two clauses, and can be written as $I \equiv P \vee Q = \big((i+1) \bmod 2 = 0 \wedge x < 2 + i + y \wedge x < 2 + y + 2i \wedge x < N + y + 1 \wedge 1 \le N\big) \vee \big(i \bmod 2 = 0 \wedge x < 2 + i + y \wedge x < 2 + y \wedge 1 \le N\big)$.

By inspecting the loop, we can derive an exact relationship between $x$, $y$, and $i$ at the beginning of each iteration. Since $x$ accumulates all even numbers less than $i$ and $y$ accumulates all odd numbers less than $i$, we obtain:

$$\text{if } i \text{ is even: } \quad x - y = -\frac{i}{2}, \qquad \text{if } i \text{ is odd: } \quad x - y = \frac{i-1}{2}.$$

Using this relationship, it is straightforward to verify that the invariant $I$ produced by UAutomizer is correct. Moreover, $I$ is strong enough to prove the final assertion $x - y \le N$.

To show that the assertion holds at loop termination, we can check the following verification condition

$$I \ \wedge \ (i = N) \ \Rightarrow \ (x - y \le N),$$

where $i = N$ denotes the loop's exit condition.

> You will be shown 3 example C programs. Please gain inspiration from the following programs to create a new high-quality C program. Do not simply copy from any of them.
>
> Requirements for the generated program:
> 1. The program MUST contain non-trivial loops (for or while).
> 2. The program MUST contain assertions.
> 3. The program MUST be compilable, self-contained, and reasonably complex (not trivial or overly short).
> 4. Only output the new C program.
>
> Example snippets:
> **Program 1**:
> `{SEED_PROGRAM_1}`
> **Program 2**:
> `{SEED_PROGRAM_2}`
> **Program 3**:
> `{SEED_PROGRAM_3}`
>
> Output format: The generated program must be wrapped strictly in the following format:
> ```c
> <NEW_C_PROGRAM>
> ```

Table A2: Prompt template for synthetic data generation.

**Case 1: $N$ is odd.** Instantiating the invariant with $i = N$ activates the $P$ disjunct of $I$, from which we obtain

$$x - y < N + 1.$$

Since $x - y$ is an integer, this directly implies $x - y \leq N$.

**Case 2: $N$ is even.** In this case, the $Q$ disjunct applies. From the clause $x < 2 + y$ contained in $Q$, we derive

$$x - y < 2.$$

Because $x - y$ is an integer and $N \geq 1$ and even (hence $N \geq 2$), we conclude

$$x - y \leq 1 \leq N,$$

establishing the desired post-condition.

Thus, the invariant $I$ indeed suffices to prove the final assertion.

**Original Program (Input to UAutomizer)**

```c
int main() {
    int N = __VERIFIER_nondet_int();
    assume_abort_if_not(N >= 1 && N < 100);
    int x = 0, y = 0;
    for (int i = 0; i < N; i++) {
        if (i % 2 == 0) {
            x += i;
        } else {
            y += i;
        }
    }
    int diff = x - y;
    __VERIFIER_assert(diff <= N);
    return 0;
}
```

**Loop Invariant Generated by UAutomizer:**

```
Line Number: 6
Predicate:
(
  (i + 1) % 2 == 0 &&
  x < 2 + i + y &&
  x < 2 + y + 2 * i &&
  x < N + y + 1 &&
  1 <= N
)
||
(
  i % 2 == 0 &&
  x < 2 + i + y &&
  x < 2 + y &&
  1 <= N
)
```

Figure A1: An example from the fine-tuning dataset: program and its loop invariant generated by UAutomizer.

