# OpenReview forum: "InvBench: Can LLMs Accelerate Program Verification with Invariant Synthesis?"
_ICLR.cc/2026/Conference — Submitted to ICLR 2026_

### Official Review · Reviewer_2Qxp · 2025-10-29

**Soundness:** 3
**Presentation:** 3
**Contribution:** 1
**Rating:** 2
**Confidence:** 4

**Summary:**

This paper presents InvBench, a benchmark and evaluation framework for assessing large language models (LLMs) on the task of loop invariant synthesis, a central challenge in program verification. Unlike prior work, which assessed LLM-generated invariants based only on correctness or similarity to ground truth (e.g., invariants generated by Daikon), InvBench introduces a verifier-based decision procedure with a formal soundness guarantee, ensuring that correctness judgments are logically valid. The benchmark evaluates both invariant correctness and verification speedup across seven state-of-the-art LLMs and several LLM-based verifiers, compared to the non-LLM solver UAutomizer.
Key results show that current LLMs struggle to produce strong invariants that meaningfully accelerate verification, though fine-tuning and Best-of-N sampling yield slight improvements.

**Strengths:**

Provides a sound and general evaluation framework for LLM-generated invariants.

Combines formal reasoning with empirical benchmarking.

Includes a large dataset derived from SV-COMP benchmarks.

**Weaknesses:**

Limited novelty.

Evaluation leads only to already established conclusions.

Dataset construction requires more detailed explanation.

Details:

The paper makes no significant novel contribution. It provides an evaluation of LLM-generated invariants in accelerating program verification and reaches the expected conclusion that LLM-generated invariants are not yet useful for verification. However, this evaluation does not yield actionable insights into how to improve LLM-based invariant synthesis.
The conclusions are largely unsurprising, reiterating well-established findings namely, that generating strong invariants is more difficult than generating correct invariants, and that model capability is a key factor in performance (lines 311–315). Line 444 also provides the takeaway that “more training is needed,” which is too vague and uninformative, especially since the experiments show that fine-tuning did not produce meaningful improvements.

The paper also claims a dataset of training instances as a central contribution, but the dataset construction process is insufficiently detailed. Lines 257–267 do not clearly explain how the dataset is created.
Given that dataset construction is a central contribution, this process should be described in more detail.

**Questions:**

How are the (899–226) seed programs expanded into 3,589 instances through LLM mutation, parameter variation, or other augmentation techniques?

If a seed program is supplied to the LLM (GPT-4o), what exactly does the LLM do—does it mutate the program to create new ones? Line 265 states that each synthetic program is analyzed by UAutomizer and only those whose assertions are proved are retained. Which assertions? The following line says that invariants extracted from UAutomizer logs are part of the fine-tuning dataset; are these the same assertions being proved by UAutomizer? How are the (899–226) seed programs expanded into 3,589 problem instances?

---

> ### Author Response · Authors · 2025-12-03
> **Response to Reviewer 2Qxp**
>
> We thank the reviewer for the insightful feedback.
>
> **Q1: How are the seed programs expanded into 3,589 problem instances?**
>
> To construct the synthetic dataset, we prompt GPT-4o using the template in Appendix A.3 (please see the revised paper). The template takes three seed programs as examples and instructs the model to synthesize a new C program that is compilable and contains both loops and assertions. To obtain a diverse and large pool of candidates, we repeatedly invoke the model with different seed programs. To avoid data leakage, these seed programs are randomly drawn from the SV-COMP pool that is disjoint from our evaluation set. Although the prompt requests compilable programs with loops and assertions, the LLM-generated programs may fail to compile, include assertions that do not hold, and contain multiple assertions. For programs with multiple assertions, we split them into separate instances, each retaining only a single assertion while preserving all loop structures (ensuring at least one loop per instance). We then run UAutomizer on every program and discard any instance that is non-compilable or whose assertion is invalid. This filtering step ensures the quality of the dataset, resulting in 3589 synthetic programs.
>
> When running UAutomizer to prove the assertions in the synthetic programs, the tool also emits the invariants it discovers. From its output, we extract the loop invariants. Each extracted invariant includes its program location and its predicate. Although each program contains exactly one assertion, it may include multiple loops, so a single program can yield multiple loop invariants, all associated with the same assertion. We pair each program with each of its corresponding loop invariants to form our training dataset. We show an example invariant generated from UAutomizer in Appendix A.4 of the revised paper.
>
> We have incorporated all the details into our revised version.
>
>
>
> **Q2: However, this evaluation does not yield actionable insights into how to improve LLM-based invariant synthesis.**
>
> We conducted a detailed breakdown analysis to understand the primary failure modes behind the lack of speedups. For each model, we categorize failures into four types:
>
> - **Incorrect invariant** — the invariant is refuted.
> - **Assume timeout** — verifying the invariant itself times out, even though UAutomizer can verify the original assertion.
> - **Assert timeout** — the invariant is verified, but verifying the final assertion *under* that invariant times out.
> - **Assume + assert timeout** — both checks time out.
>
> The tables below summarize this breakdown for the top three models on the easy split and the strongest-performing model on the hard split, showing the number of instances falling into each failure mode.
>
> ### **Easy Split Failure Breakdown**
>
> | Model | Incorrect | Assume Timeout | Assert Timeout | Assume + Assert Timeout |
> |-------|-----------|----------------|----------------|--------------------------|
> | **claude-opus-4.1** | 10 | 13 | 22 | 58 |
> | **o3** | 29 | 17 | 19 | 17 |
> | **gpt-5** | 20 | 4 | 18 | 65 |
>
> ### **Hard Split Failure Breakdown**
>
> | Model (Hard Split) | Incorrect | Assume Timeout | Assert Timeout | Assume + Assert Timeout |
> |---------------------|-----------|----------------|----------------|--------------------------|
> | **gpt-oss-120b** | 10 | 12 | 32 | 58 |
>
> This breakdown highlights why LLMs rarely produce speedups. While a portion of failures stem from incorrect invariants, the more fundamental issue is that the invariants generated by LLMs rarely decompose the verification task into strictly easier subgoals. As a result, both the invariant-checking query and the assertion-under-invariant query frequently time out, as reflected in the large number of cases in the “Assume + Assert Timeout’’ category. This suggests that current models lack an understanding of what makes a verification query easy or difficult for symbolic solvers.
>
> Future work should explore strategies that help models internalize or predict solver difficulty, such as training reward models, so that they can propose invariants that genuinely simplify the verification task rather than inadvertently increasing solver burden. We have incorporated this analysis into the revised paper.
>
>
> **“The experiments show that fine-tuning did not produce meaningful improvements.”**
>
> We respectfully disagree. The creation and release of our fine-tuning dataset is a key contribution of this work, and fine-tuning yields substantial improvements in practice. For example, fine-tuning Qwen3-Coder-480B increases the percentage of correct invariants from 14.2% to 40.7%, and the percentage of speedup cases from 8% to 29.2%. These results demonstrate that supervised fine-tuning can meaningfully and significantly enhance LLM performance on invariant synthesis.

---

### Official Review · Reviewer_ZjxM · 2025-10-31

**Soundness:** 1
**Presentation:** 2
**Contribution:** 1
**Rating:** 2
**Confidence:** 4

**Summary:**

This paper introduces InvBench, a benchmark to measure the capabilites of LLMs for generating invariants that accelerate the program verification process.

The authors claim that prior benchmarks using LLMs (or LLM-based systems) for invariant synthesis (a) use customized datasets and (b) omit comparisons with state-of-the-art symbolic baselines. For the generated invariants to be useful, they need to be weak enough to be provable and strong enough to help prove the target assertion. To measure these, the InvBench benchmark measures the correctness of the generated invariants and the speedup that they provide for verification as compared to the UAtomizer symbolic solver. An evaluation of 7 LLMs and prior LLM-based systems shows that, for the most part, given the same timeout (with the time for LLM-based systems including inference time), UAtomizer outperforms LLM-based methods. Finally, the paper shows that fine-tuning and best-of-N sampling can improve the performance of LLMs on InvBench.

**Strengths:**

- The paper provides a unified evaluation of multiple LLMs and LLM-based systems on invariant synthesis, which, even though I have reservations about the setup, provides a useful picture of where the field stands today.

**Weaknesses:**

- The claim that prior works omit comparisons with state-of-the-art symbolic baselines is false. In particular, the authors claim in multiple places in the paper, in the Introduction and Results sections, that Loopy [1] [2] (for some reason, they cite the newer version in the introduction but compare against the older version in Section 5) does not compare against UAtomizer. But both the papers do compare against UAtomizer, and even report that UAtomizer outperforms Loopy at invariant synthesis. This raises serious concerns about the veracity of the claims made in the paper.
- The above point also means that the core insight of the paper (that UAtomizer is still better than LLMs) has been reported before, calling into question the novelty of the work.

- It is unclear why InvBench is different/more useful than the other "customized datasets" that the paper mentions. The benchmarks used in [2] and [3] both draw from the benchmark used by Code2Inv [4], SV-COMP problems and some other sources. InvBench also draws its problems from SV-COMP, and thus seems quite similar to prior benchmarks.

- The paper makes two design choices for evaluation, whose justification is unclear to me. First, it measures correctness of generated invariants and speedup due to generated invariants on benchmarks that a symbolic solver (i.e., UAtomizer) can already solve. In contrast, prior works (e.g., [2]) measure whether adding the generated invariants as annotations allow a symbolic solver to verify the program. Being able to verify programs that a solver could not have solved without the generated invariants seems closer to the goal of invariant synthesis than speeding up the verification of programs that a symbolic solver can already verify.

- Second, it includes the LLM inference time in the timeout measurement when comparing LLMs and LLM-based systems to UAtomizer. I understand that this is to compare the overall verification time between LLM (or LLM-based system) + UAtomizer and UAtomizer. However, given the variability of inference times (because they involve network calls to model providers), it is unclear whether the results are due to capability of LLMs or just the slowness of APIs. Running experiments with a larger timeout (which would make the inference time a smaller portion of the total time) seems like a more useful experiment.  I believe that is quite relevant because the UAtomizer solving time can scale very quickly for small but complex programs while the LLM inference time would remain roughly the same. Thus, if total time is what we are interested in, then it is important understand how it scales for complex programs.

[1]: Kamath, Adharsh, et al. "Finding inductive loop invariants using large language models." arXiv preprint arXiv:2311.07948 (2023).

[2]: A. Kamath et al., "Leveraging LLMs for Program Verification," 2024 Formal Methods in Computer-Aided Design (FMCAD).

[3]: Guangyuan Wu, Weining Cao, Yuan Yao, Hengfeng Wei, Taolue Chen, and Xiaoxing Ma. 2024. LLM Meets Bounded Model Checking: Neuro-symbolic Loop Invariant Inference. In Proceedings of the 39th IEEE/ACM International Conference on Automated Software Engineering (ASE '24).

[4]: Xujie Si, Aaditya Naik, Hanjun Dai, Mayur Naik, and Le Song. 2020. Code2Inv: A Deep Learning Framework for Program Verification. In Computer Aided Verification: 32nd International Conference, CAV 2020, Los Angeles, CA, USA, July 21–24, 2020, Proceedings, Part II.

**Questions:**

1. Could you clarify the point about previous approaches not comparing with symbolic tools, and, in particular, Loopy not comparing with UAtomizer?

2. Given that the Loopy paper already shows that UAtomizer outperforms LLMs on invariant synthesis, what is/are the new insight(s) provided by your experiments?

3. How is InvBench different and/or more useful than the benchmarks used in Loopy and LaM4Inv?

4. Could you explain the rationale for the two design choices I have mentioned above?

5. For the second one, is it possible to run an experiment with a larger timeout? I believe that would be quite relevant for the reasons I stated.

---

> ### Author Response · Authors · 2025-12-03
> **Response1 to Reviewer ZjxM**
>
> We thank the reviewer for the constructive feedback.
>
>
> **Q1: Loopy papers do report comparisons showing UAutomizer outperforming Loopy. Could the authors clarify what they mean by saying Loopy does not compare with UAutomizer?**
>
>
> Thank you for pointing this out. We agree that Loopy reports results for comparison with UAutomizer. Our concern is that the comparison is not performed fairly. Specifically, Loopy does not run UAutomizer and its own method under the same timeout budget: Loopy stops after issuing 15 LLM queries, whereas UAutomizer is not subject to an equivalent time constraint. By contrast, our evaluation enforces identical timeouts for all tools and includes LLM inference time to ensure fairness.
>
> Moreover, the comparison between Loopy and LEMUR in the Loopy paper is also difficult to interpret: the two tools are compared using the same number of LLM queries rather than the same wall-clock time. As a result, the claim in the Loopy paper that Loopy performs comparably to LEMUR is not well supported.
>
> These limitations in prior work further motivate our systematic evaluation, where all LLM-based verifiers and UAutomizer are run on the same dataset under identical timeout budgets, providing a fair and direct comparison.
>
>
>
> **Q2: Given that the Loopy paper already shows that UAtomizer outperforms LLMs on invariant synthesis, what is/are the new insight(s) provided by your experiments?**
>
> Our experiments highlight a key insight for the community: the effectiveness of LLM-based verifiers is strongly determined by the underlying symbolic solver they rely on. This conclusion does not follow from a single comparison (e.g., Loopy vs. UAutomizer) but emerges only when examining multiple systems built on different solvers: LaM4Inv uses ESBMC, Loopy uses Frama-C, and LEMUR uses UAutomizer.
>
> Across these works, we find that LLM-based verifiers rarely outperform the strongest symbolic solver unless they are already built on top of it. Although each system reports gains relative to its own underlying solver on its own dataset (with the exception that Frama-C cannot prove assertions without Loopy’s invariants), these improvements do not necessarily translate when compared against a state-of-the-art solver under a uniform evaluation protocol. LEMUR’s comparatively strong performance appears closely tied to its use of UAutomizer rather than to the LLM component alone.
>
> This broader perspective is the new insight our work provides beyond prior isolated comparisons.
>
>
>
> **Q3: How is InvBench different and/or more useful than the benchmarks used in Loopy and LaM4Inv?**
>
> We investigate the distributional differences between datasets. Programs in InvBench are significantly longer, averaging 62 lines of code in the hard split, compared to only 22 for LaM4Inv, 23 for LEMUR, and 27 for Loopy. Manual inspection shows that InvBench contains features such as multiple loops, functions, arrays, and pointers, which are largely absent from prior datasets. This makes InvBench a more challenging and realistic setting that better differentiates solver capabilities. In our evaluation, systems such as LaM4Inv perform well only on their own curated datasets but fail to generalize to InvBench, raising concerns about overfitting and the robustness of prior techniques.
>
> Furthermore, InvBench releases the first large-scale open-source training corpus with verifier-generated ground-truth invariants, enabling substantial gains through supervised fine-tuning.
>
> Beyond the dataset itself, we introduce a simple, verifier-based decision procedure for program verification. To our knowledge, InvBench is the first to enable a sound evaluation of LLM-generated invariants across different solvers.
>
> Finally, our method establishes a new performance baseline. Despite its simplicity, the decision procedure enables LLMs to outperform UAutomizer, whereas prior LLM-based verifiers with significantly more complex designs rarely do so. Moreover, basic techniques such as supervised fine-tuning and Best-of-N sampling yield substantial gains, resulting in a strong new state-of-the-art baseline built on simple yet effective methods.

---

> ### Author Response · Authors · 2025-12-03
> **Response2 to Reviewer ZjxM**
>
> **Q4: Prior work evaluates whether adding LLM-generated invariants enables a solver to verify programs it could not solve before. In contrast, your evaluation measures invariant correctness and speedup on programs that UAutomizer already solves. Why do you focus on accelerating solvable cases?**
>
> We address the question from two perspectives.
>
> **Solver capability.** Prior work Loopy builds on Frama-C, a solver that does not discover invariants on its own; it can only verify programs when supplied with invariants (e.g., from an LLM). In contrast, many modern verifiers, including UAutomizer, have their own internal invariant-generation mechanisms and can already verify programs without any external annotations. Our methodology is designed for this latter setting. Since UAutomizer represents the current state of the art, our evaluation assumes a solver that is capable of verification on its own, and the question becomes whether LLM-generated invariants can improve its behavior.
>
> **Timing and speedup.** Our evaluation does demonstrate cases where LLM-generated invariants enable verification within a given timeout when UAutomizer alone would time out. To quantify improvements systematically, however, we focus on instances that UAutomizer solves within 600 seconds. Measuring speedup on solvable instances allows us to compare models fairly and consistently. Moreover, any instance that exhibits a speedup necessarily implies that for some smaller timeout threshold, the solver with LLM-generated invariants can succeed while the baseline solver would fail, capturing the “verify previously unsolved programs” scenario implicitly.
>
>
>
> **Q5: The evaluation includes LLM inference time. Did you run experiments with sufficiently large timeouts so that inference time becomes only a small fraction of the total verification time?**
>
> We report here the ratio of LLM inference time to total verification time. On the easy split, this ratio is **16%**, and on the hard split, it drops further to **5%**, since verification itself dominates the runtime.
>
> Unlike Loopy, which invokes the LLM up to 15 times and therefore spends a substantial portion of its time budget on API calls, our evaluation only calls the LLM once per program. As a result, **LLM inference accounts for only a small fraction of total runtime**. Moreover, the hard split contains sufficiently challenging programs that solver time naturally outweighs inference overhead, making the comparison meaningful even when inference latency is included.

---

### Official Review · Reviewer_GZ5Z · 2025-11-01

**Soundness:** 3
**Presentation:** 3
**Contribution:** 2
**Rating:** 6
**Confidence:** 2

**Summary:**

This work introduces a method for evaluating alleged loop invariants, in particular loop invariants generated by LLMs. Rather than comparing to a known ground-truth or "gold standard" invariant synthesis tool, the method proposed works as follows. Consider a property $p*$ (e.g. some assertion) and the loop while($C$){do $B$; assert $p^*$}.
- Suppose that the LLM proposes the invariant $I$ holds at the start of each loop. That is, while($C$){do assert $I$; $B$}.
- Then, use an external tool to prove two claims: First, while($C$){do assert $I$; $B$}, and second while($C$){do assume $I$; $B$; assert $p^*$}.

If a batch of invariants is generated, the method is applied individually to each. Only invariants proved by the external tool are retained.

The authors use this method to generate a dataset, InvBench, as follows. Starting from SV-COMP (899 instances), use UAutomizer to obtain 3589 program-invariant pairs. The authors then test several LLM-based invariant generators on the benchmark. Fine-grained analysis depending on the instance difficulty (measured by UAutomizer running time) is presented. Runtime and success rate are the main metrics, i.e., how long does it take for a server-hosted LLM to generate a correct invariant, including latency introduced by the server. Additional experiments show that fine-tuning on a train split of the dataset improves performance on a test split.

**Strengths:**

- Loop invariant generation is an important problem, and a promising domain for LLM-based improvements upon classical tools.
- The method underlying InvBench data generation is simple and easy to understand. I emphasize that simplicity is a strength, not a weakness, of scientific work.
- The method is proven to be sound.
- In recent years, fine-tuning has largerly been abandoned in favor of in-context (or zero-shot) methods in many domains. Thus, showcasing a domain in which LLMs showcase poor zero-shot performance which is dramatically improved by fine-tuning, is significant.
- It's tricky to write an "interdisciplinary" paper such as this, at the intersection of ML and formal verification. Although I consider myself an outsider to the latter, I was still mostly able to follow the approach and meaning of experiments. The figure and concrete examples helped.

**Weaknesses:**

- If my summary above is correct, it seems that the finetuning approach is essentially Imitation Learning on UAutomizer. As such, comparing the neural methods to UAutomizer itself seems strange. Concretely, consider scaling up this approach (e.g. as a way to get new SOTA invariant generators). Is the idea that given enough data, eventually a fine-tuned LLM would be able to replace UAutomizer (either it discovers more invariants, or does so faster)?
- In-context learning (k-shot prompting) is not examined. I don't expect it to do much better than zero-shot, but considering its ubiquity it should be tested.
- In general, the significance of this new dataset (and approach) would be strengthened if a better experimental analysis was carried out. Right now, the analysis mainly examines whether existing LLMs are able to achieve speed-up on UAutomizer, and the answer is generally "no". It would be interesting to conduct an error analysis to understand _why_ these tools fail. Is there an inherent limitation to LLM-based invariant synthesis, or is it a matter of generalizability (e.g. LLMs do very well of instances of a certain type, and poorly on others)? Considering that papers showing that LLMs _can't_ do a certain task are exceedingly rare, I think changing the narrative (and supporting this experimentally) to "LLMs fail on invariant synthesis, and here is a concrete hill to climb on (a new dataset)" would be compelling.

**Questions:**

Statements in my summary and Weaknesses can be taken as questions, and I welcome comments by the authors.

---

> ### Author Response · Authors · 2025-12-03
> **Response to Reviewer GZ5Z**
>
> We thank the reviewer for the thoughtful feedback.
>
> **Q1: Is the idea that given enough data, eventually a fine-tuned LLM would be able to replace UAutomizer (either it discovers more invariants, or does so faster)?**
>
> Our approach is not intended to replace UAutomizer. It is inherently neuro-symbolic: the LLM proposes candidate invariants, and a sound symbolic verifier (UAutomizer) checks both their correctness and whether they suffice to prove the final assertion. While UAutomizer can already generate invariants on its own, fine-tuning helps the LLM propose stronger invariants, but it does not eliminate the need for a verifier. Even with extensive training, an LLM cannot provide the soundness guarantees required in program verification, so the symbolic component remains essential. Given this setup, comparing our neuro-symbolic pipeline (LLM + UAutomizer) to UAutomizer alone is appropriate. UAutomizer is the verifier that provides soundness guarantee, and the goal is to determine whether LLM-generated invariants can accelerate UAutomizer’s verification process, not to replace it.
>
>
> **Q2: In-context learning (k-shot prompting) is not examined. I don't expect it to do much better than zero-shot, but considering its ubiquity, it should be tested.**
>
> Thank you for the suggestion. We conducted additional experiments using **0-shot, 2-shot, and 4-shot** prompting. The results show that *in-context learning provides only marginal gains* and does not change the overall conclusion: strong invariants remain challenging for current models, and performance improvements are limited compared to zero-shot prompting.
>
> The results on the easy split are summarized below.
>
> | Model | Shots | % Correct Invariant | % Speedup | Speedup_{>1} | Speedup_{all} |
> |-------|--------|----------------------|------------|----------------|-----------------|
> | **gpt-5** | 0 | 37.2% | 26.5% | 1.24× | 1.06× |
> |          | 2 | 35.4% | 24.8% | 1.26× | 1.05× |
> |          | 4 | 36.3% | 30.1% | 1.31× | 1.10× |
> | **o3**   | 0 | 39.8% | 28.3% | 1.37× | 1.09× |
> |          | 2 | 39.8% | 29.2% | 1.39× | 1.10× |
> |          | 4 | 39.8% | 30.0% | 1.41× | 1.12× |
> | **claude-opus-4.1** | 0 | 18.6% | 8.8% | 1.23× | 1.02× |
> |                     | 2 | 18.6% | 8.8% | 1.25× | 1.02× |
> |                     | 4 | 18.6% | 8.8% | 1.27× | 1.04× |
>
>
>
> **Q3: It would be interesting to conduct an error analysis to understand why LLMs fail to achieve speedup on UAutomizer.**
>
> We conducted a detailed breakdown analysis to understand the primary failure modes behind the lack of speedups. For each model, we categorize failures into four types:
>
> - **Incorrect invariant** — the invariant is refuted.
> - **Assume timeout** — verifying the invariant itself times out, even though UAutomizer can verify the original assertion.
> - **Assert timeout** — the invariant is verified, but verifying the final assertion *under* that invariant times out.
> - **Assume + assert timeout** — both checks time out.
>
> The tables below summarize this breakdown for the top three models on the easy split and the strongest-performing model on the hard split, showing the number of instances falling into each failure mode.
>
> ### **Easy Split Failure Breakdown**
>
> | Model | Incorrect | Assume Timeout | Assert Timeout | Assume + Assert Timeout |
> |-------|-----------|----------------|----------------|--------------------------|
> | **claude-opus-4.1** | 10 | 13 | 22 | 58 |
> | **o3** | 29 | 17 | 19 | 17 |
> | **gpt-5** | 20 | 4 | 18 | 65 |
>
> ### **Hard Split Failure Breakdown**
>
> | Model (Hard Split) | Incorrect | Assume Timeout | Assert Timeout | Assume + Assert Timeout |
> |---------------------|-----------|----------------|----------------|--------------------------|
> | **gpt-oss-120b** | 10 | 12 | 32 | 58 |
>
> This breakdown highlights why LLMs rarely produce speedups. While a portion of failures stem from incorrect invariants, the more fundamental issue is that the invariants generated by LLMs rarely decompose the verification task into strictly easier subgoals. As a result, both the invariant-checking query and the assertion-under-invariant query frequently time out, as reflected in the large number of cases in the “Assume + Assert Timeout’’ category. This suggests that **current models lack an understanding of what makes a verification query easy or difficult for symbolic solvers**.
>
> Future work should explore strategies that **help models internalize or predict solver difficulty, such as training reward models**, so that they can propose invariants that genuinely simplify the verification task rather than inadvertently increasing solver burden. We have incorporated this analysis into the revised paper.

---

### Official Review · Reviewer_x7S4 · 2025-11-02

**Soundness:** 3
**Presentation:** 3
**Contribution:** 2
**Rating:** 6
**Confidence:** 3

**Summary:**

Strong loop invariants are important for efficient verification. This paper presents InvBench, a simple formal framework and benchmark for evaluating the ability of LLMs to synthesise loop invariants. Prior work on this by Pei et al (2023) has several shortcomings, including no consideration of the strength of generated invariants, and a notion of "correctness" based on (syntactic?) comparison of generated invariants to those generated by another (unsound by design) tool. And just naively comparing generated invariants (presumably syntactically) fails to detect invariants that are semantically equivalent.

The formal framework part of InvBench improves on this using a two-step interaction with an external verifier V (assumed to be sound) to evaluate a candidate invariant q. Given target property p* and program P, the two steps are (if my understanding is correct):

- V(P, ∅, q) to check if q is actually an invariant (i.e. holds for all executions)
- V(P, {q}, p*) to check if the target property holds when q is treated as an assumption (i.e. holds for all  executions where q is satisfied).

The first step checks q for correctness and the second step then allows q to be tested for speedup in the verification of p*.

The authors also highlight another strand of related work (building on Pei et al) which embeds LLMs into verification frameworks. This presumably is closer to the work presented here, but this prior work suffers from a lack of standard benchmarks for evaluation, and lack of comparison against a traditional verifier baseline (e.g. Ultimate Automizer =  UAutomizer). A second contribution of the paper is thus an invariant synthesis benchmark which is used to comparatively evaluate some of the prior work. Overall the LLM-based approaches perform slightly worse than UAutomizer but the paper also shows that fine-tuning and Best-of-N sampling improve things a bit.

**Strengths:**

- It's nice to have the formal framework which uses soundness rather than relying on "equivalence" (syntactic or otherwise) to a system which isn't sound. The framework itself is simple, parameterised by an external verifier acting as an oracle. Proof of soundness of the decision procedure seems trivial (not necessarily a weakness though).
- Makes a convincing case that a standard benchmark for invariant synthesis would be useful and reports on using such a benchmark to evaluate a bunch of existing LLM-based approaches.
- Provides some evidence that current LLMs find the invariant synthesis problem hard.

**Weaknesses:**

- It's nice that the framework itself is parameterised on V. However the heavy reliance on UAutomizer as the baseline solver for the experimental results (e.g. determining the split of the dataset and speedup) might limit the generality of the results.

- The benchmark itself seems to be a specific (randomly chosen) subset of problems from SV-COMP, after partitioning into easy/hard according to UAutomizer. The benchmark itself is described as a contribution, but the benchmark itself (or sufficient metadata to reconstruct it) wasn't submitted as Supplementary Material.

**Questions:**

### Questions

- Are there reasons other traditional verifiers beyond UAutomizer haven't been considered? If an advantage of your framework is that V is "pluggable", perhaps could that be demonstrated by trying out other (sound) verifiers.

- Table 2 and elsewhere: for the point estimates given for correctness/speedup, was there a reason not to report standard deviations or some other indication of uncertainty? And do the figures reported represent single runs or averages over multiple runs of a given model?

- Do you plan to release your benchmark problem set (and perhaps other derivatives of SV-COMP based on verifiers other than UAutomizer)?

### Minor suggestions

Minor textual suggestions that didn't impacted my score but could be improved during the revision/discussion
phase:

- l.80 (and elsewhere): Is "customized" the right word here? "Customized" implies some modification of an existing artifact, whereas perhaps you mean a unique dataset specific to that solution? If so, a better choice of word might be "custom" (or "bespoke", or "idiosyncratic").
- l.82: "underscoring": Not sure this is what you mean. (The fact that earlier work omits such comparisons doesn't highlight the need for them -- that's what you're doing!) Maybe you can just omit this clause.
- l.103: "the only exception" -> "the exception"
- l.151: Is it correct to say "Loopy...*builds*" an LLM-based pipeline, rather than "Loopy...*is*"?
- l.160: Citation parentheses need fixing for (Hoare, 1969).
- l.211: d /neq F might be fractionally clearer than d \in {T, U}.
- l.266: "UAutomizer 's" has a spurious space. You might be able to fix that with \xspace (if you're using a
  macro).
- l.427: LoRA needs a citation.
- l.444: "more training efforts are required" -> "more training effort is required"

- The summary of contributions (l.118 onwards) is a bit redundant with the preceding 2 paragraphs. I would think about combining -- i.e. taking the earlier two paragraphs, streamlining slightly, and then making those the summary of contributions 2 and 3. That will make for a smoother read. The contributions could also forward-ref the relevant  sections, so they also work as a summary of the paper.

- Second paragraph of Related Work, especially the second half of the paragraph, is rather cursory. From Chakraborty to Liu there isn't really any commentary, so it's hard to understand the relevance. Maybe just a short sentence to introduce them all as different learning-based methods for improving LLM synthesis would help? And Liu feels a bit out of place; it's just a dataset. Taking a bit more time over this paragraph would be good.

---

> ### Author Response · Authors · 2025-12-03
> **Response to Reviewer x7S4**
>
> We thank the reviewer for the insightful feedback.
>
> **Q1: Are there reasons other traditional verifiers beyond UAutomizer haven't been considered? If an advantage of your framework is that V is "pluggable", perhaps could that be demonstrated by trying out other (sound) verifiers.**
>
> We chose UAutomizer because it is the current state-of-the-art solver in the international competition of SV-COMP 2025 and therefore serves as the strongest and most widely accepted symbolic baseline. Our decision procedure, however, does not rely on any feature specific to UAutomizer. It only requires that the verifier be sound on conclusive outcomes, which is a standard property of modern symbolic verifiers. As a result, the framework is compatible with other solvers.
>
> To illustrate this flexibility, we also plugged in ESBMC, another commonly used verifier. When paired with the o3 model, ESBMC can prove 8.8% of the invariants that o3 generates within the time limit, and achieve speedups on 1.8% percent of instances, with a conditional speedup of 1.27x, and an overall speedup of 1.004x. This demonstrates the flexibility that our framework provides.
>
>
> **Q2: Table 2 and elsewhere: for the point estimates given for correctness/speedup, was there a reason not to report standard deviations or some other indication of uncertainty? And do the figures reported represent single runs or averages over multiple runs of a given model?**
>
> We agree that reporting uncertainty is important. We are running the full end-to-end evaluation five times and will report 95% confidence intervals for all metrics in the revision. Below we show results for three models on the easy split. The narrow intervals suggest that the measurements are statistically stable.
>
>
> | Model               | % Correct Invariant     | % Speedup            | Speedup_{>1}         | Speedup_{all}        |
> |---------------------|--------------------------|-----------------------|------------------------|------------------------|
> | **claude-sonnet-4** | 18.4% ± 2.0%             | 8.8% ± 1.4%           | 1.24× ± 0.10          | 1.02× ± 0.01          |
> | **gpt-5**           | 37.7% ± 2.2%             | 26.5% ± 3.9%          | 1.24× ± 0.05          | 1.06× ± 0.02          |
> | **o3**              | 39.8% ± 1.6%             | 28.3% ± 3.6%          | 1.37× ± 0.04          | 1.09× ± 0.02          |
>
>
>
> **Q3: Do you plan to release your benchmark problem set (and perhaps other derivatives of SV-COMP based on verifiers other than UAutomizer)?**
>
> Yes. We have released the benchmark problem set and the full training dataset on GitHub. For the review process, we provide an anonymized version at https://anonymous.4open.science/r/InvBench/
>
> We are also cleaning up the codebase to support both UAutomizer and additional solvers, such as ESBMC, and we will include our full script in the public release in the revision.
>
>
> **Q4: Writing suggestions**
>
> We sincerely thank the reviewer for the detailed writing suggestions. We have incorporated all the suggestions into the revised version.

---

### Author Response · Authors · 2025-12-03
**Rebuttal Summary**

## Rebuttal Summary for Area Chairs and Reviewers

Because author-reviewer interactions were disabled, we briefly summarize the key concerns raised and how we addressed them.

### **Verifier choice (Reviewer x7S4)**
The reviewer asked why only UAutomizer was used, whether the framework is truly “pluggable”.

→ We clarified that UAutomizer is the SV-COMP 2025 state-of-the-art baseline.

→ We demonstrated flexibility by integrating **ESBMC** and reporting its results.

### **Uncertainty and statistical robustness (Reviewer x7S4)**
The reviewer requested confidence intervals.

→ We ran **five independent evaluations** and now report **95% confidence intervals**, which are consistently narrow.

### **Dataset release and more details on synthetic dataset construction (Reviewer x7S4, Reviewer 2Qxp)**
Reviewers asked about the benchmark release and dataset generation.

→ We released the dataset for both training and evaluation.

→ We added a detailed explanation for the synthetic dataset generation.

### **In-context learning (Reviewer GZ5Z)**
The reviewer requested k-shot prompting experiments.
→ We added **0-shot, 2-shot, and 4-shot** evaluations.

→ Results show only marginal gains, confirming that in-context prompting does not change conclusions.

### **Failure Analysis and Actionable insights for improving LLM-based invariant synthesis (Reviewers GZ5Z, 2Qxp)**
The reviewer asked for failure analysis and whether the evaluation yields guidance on improving LLM-generated invariants.

→ We added a detailed **failure-mode analysis** for why LLM-generated invariants do not provide speedup.

→ We detailed failure mode breakdown shows that **current models lack an understanding of what makes a verification query easy or difficult for symbolic solvers**.

→ We suggest future research should explore strategies that **help models internalize or predict solver difficulty, such as training reward models**.


### **Motivation for Comparison and new insights beyond prior work (Reviewer ZjxM)**
The reviewer questions our motivation for comparison and what new findings our experiments provide.

→ We clarified fairness issues in prior work’s comparison (e.g., Loopy does not use uniform timeouts), whereas our evaluation uses identical timeouts.

→ We highlight a key insight for the community: **the effectiveness of LLM-based verifiers is strongly determined by the underlying symbolic solver they rely on** (ESBMC vs. Frama-C vs. UAutomizer). This is the new insight our work provides beyond prior isolated comparisons.

→ Our method results in **a strong new state-of-the-art baseline built on simple yet effective methods**:  our decision procedure, combined with supervised fine-tuning and Best-of-N sampling, enables LLMs to outperform UAutomizer, whereas prior LLM-based verifiers with significantly more complex designs rarely do so.

### **Clarification on Effectiveness of fine-tuning (Reviewer 2Qxp)**
The reviewer interpreted the results as showing limited improvements from fine-tuning, which we clarify is not the case.

→ We report substantial gains (e.g., Qwen3-Coder-480B: **14.2% → 40.7%** correctness, **8% → 29.2%** speedups).

---

In summary, we added new experiments (ESBMC integration, k-shot prompting, 5-run confidence intervals, failure-mode analysis), clarified dataset construction, and provided new insights and clarifications, directly addressing all reviewer questions. We have incorporated the changes into the revised version (with changes marked in red).

Thank you again for the time and care devoted to reviewing our submission.

---

### Meta-Review · Area_Chair_weHC · 2026-01-06

**Summary:**

The paper proposes a new framework for evaluating the ability of LLMs to synthesize loop invariants. Reviewers recognize that the framework is formal and unified across multiple LLMs and LLM-based systems, and that the released fine-tuning dataset significantly improves performance.

The main concerns relate to the limited contribution and lack of new insights. Reviewers 3 and 4 argue that the conclusions do not go beyond existing findings, as prior work has already shown that LLM-based methods do not outperform symbolic verifiers. Reviewer 3 also questions the strength of the proposed InvBench and the evaluation settings, including the use of speedup and timeout metrics. Reviewer 4 notes that the analysis does not provide actionable guidance on how to improve LLM-based invariant synthesis.

The authors’ rebuttal addresses several points, such as dataset construction details, in-context learning results, and the difference between InvBench and prior works. However, it does not resolve the core concerns regarding the lack of new insights and the strength of the evaluation metrics. In particular, the observation that performance is largely determined by the underlying symbolic solver is unsurprising, and the discussion on solver capability does not adequately account for cases such as LEMUR, which also uses UAutomizer.

Given the shared and major concerns raised by two reviewers who recommend rejection, I lean toward rejection.

**Reviewer Concerns:**

Reviewer 3’s concern about the contribution of the benchmark is likely addressed, and Reviewer 4’s concern about the effectiveness of the training data is also addressed.

However, the concerns about the evaluation settings and the lack of new insights are unlikely to be resolved.

**Reviewer Scores:**

Reviewers 1 and 2 may not change their scores. Although their concerns are largely addressed, they have relatively low confidence ratings.

Reviewers 3 and 4 are also unlikely to change their scores, as their main concerns are not fully addressed.

---

### Decision · Program_Chairs · 2026-01-26

Reject